# Computation of the Hausdorff Distance between Two Compact Convex Sets

Kenneth Lange [1,2,3]

1 Department of Computational Medicine, University of California, Los Angeles, CA 90095, USA; klange@ucla.edu; Tel.: +1-310-206-8076
2 Department of Human Genetics, University of California, Los Angeles, CA 90095, USA
3 Department of Statistics, University of California, Los Angeles, CA 90095, USA

**Abstract:** The Hausdorff distance between two closed sets has important theoretical and practical applications. Yet apart from finite point clouds, there appear to be no generic algorithms for computing this quantity. Because many infinite sets are defined by algebraic equalities and inequalities, this a huge gap. The current paper constructs Frank–Wolfe and projected gradient ascent algorithms for computing the Hausdorff distance between two compact convex sets. Although these algorithms are guaranteed to go uphill, they can become trapped by local maxima. To avoid this defect, we investigate a homotopy method that gradually deforms two balls into the two target sets. The Frank–Wolfe and projected gradient algorithms are tested on two pairs $(A, B)$ of compact convex sets, where: (1) $A$ is the box $[-\mathbf{1}, \mathbf{1}]$ translated by $\mathbf{1}$ and $B$ is the intersection of the unit ball and the non-negative orthant; and (2) $A$ is the probability simplex and $B$ is the $\ell_1$ unit ball translated by $\mathbf{1}$. For problem (2), we find the Hausdorff distance analytically. Projected gradient ascent is more reliable than the Frank–Wolfe algorithm and finds the exact solution of problem (2). Homotopy improves the performance of both algorithms when the exact solution is unknown or unattained.

**Keywords:** convex set; distance; Frank–Wolfe; projected gradient





## 1. Introduction

The Hausdorff distance [1,2] between two compact sets $A$ and $B$ in a Euclidean space $\mathbb{R}^p$ is defined as

$$d_H(A, B) \;=\; \max\{d(A, B), d(B, A)\},$$

where

$$d(A, B) \;=\; \max_{\boldsymbol{x} \in A} \operatorname{dist}(\boldsymbol{x}, B), \quad d(B, A) \;=\; \max_{\boldsymbol{x} \in B} \operatorname{dist}(\boldsymbol{x}, A),$$

$\operatorname{dist}(\boldsymbol{x}, A) = \min_{\boldsymbol{y} \in A} \|\boldsymbol{x} - \boldsymbol{y}\|$, and $\operatorname{dist}(\boldsymbol{x}, B) = \min_{\boldsymbol{y} \in B} \|\boldsymbol{x} - \boldsymbol{y}\|$. Here, $\|\cdot\|$ denotes the standard Euclidean norm in $\mathbb{R}^p$. The Blaschke formula

$$d_H(A, B) \;=\; \max_{\boldsymbol{x}} |\operatorname{dist}(\boldsymbol{x}, A) - \operatorname{dist}(\boldsymbol{x}, B)|$$

serves as an alternative definition of Hausdorff distance [3]. Wikipedia has a helpful entry on Hausdorff distance with a two-dimensional illustration. The theoretical value of Hausdorff distance stems from the fact that it turns the collection of compact sets into a complete separable metric space. In general, Hausdorff distance is challenging to compute.

Hausdorff distance has many applications. For instance, it is instrumental in defining continuity, compactness, and completeness for integral operators, differential operators, and Fourier transforms in functional analysis [4,5]. These concepts are in turn relevant

to the analysis of the existence, uniqueness, and stability of solutions to various equations in mathematics and physics [6]. In computer vision, Hausdorff distance enables object recognition [7] and allows one to quantify the difference between two different representations of the same object [8]. Edge detection and pixelization are usually necessary preprocessing steps. Other applications include robotics [9], the fractal modeling of biological structures [10], and the numerical computation of attractors in dynamical systems [11].

The current paper derives and tests two new algorithms for computing the distance $d_H(A, B)$ between two compact convex sets. The previous work on this intrinsically interesting problem is mostly limited to finite point clouds, usually in two and three dimensions [12–16]. The formulas

$$d(A, B) \quad = \quad \max_{x \in A} \min_{y \in B} \|x - y\| \quad \text{and} \quad d(B, A) \quad = \quad \max_{x \in B} \min_{y \in A} \|x - y\|$$

can be naively implemented for finite sets $A$ and $B$. The naive implementation benefits from fast software such as the Julia Distances package, which exploits matrix multiplication to find all Euclidean distances between the column vectors of two matrices. The ImageDistances.jl package (github.com/JuliaImages/Images.jl) appears to rely on the naive method [12] for computing Hausdorff distances. Once the distances $d_{ij} = \|x_i - y_j\|$ are computed, the computational complexity of the naive method is $O(mn)$, where $m$ and $n$ equal the number of points $x_i \in A$ and $y_j \in B$, respectively. This complexity can be reduced by various devices, as suggested in the cited references.

The algorithms for computing the Hausdorff distance between two polygons [17], two curves [18] in the plane, and a curve and a surface [19] represent exceptions to the discrete point-cloud method. More complicated sets defined by algebraic formulas can be attacked by pixelating the sets or sampling them at a dense set of random points. The methods of continuous optimization offer an attractive alternative to the various current methods. Although the calculation of $d_H(A, B)$ takes us outside the comfortable realm of convex optimization, the tools of convex calculus are highly pertinent. To their credit, these tools perform well in higher dimensions. It remains to be seen whether the Hausdorff distance will have practical value in shape recognition in this regime. It would be prudent to keep the possibility in mind.

To calculate $d_H(A, B)$, it clearly suffices to calculate $d(A, B)$ and $d(B, A)$ separately and take the maximum. For many sets $B$, dist$(x, B)$ is explicitly known or can be computed by an efficient algorithm [20,21]. The Euclidean distance dist$(x, B)$ can be expressed as

$$\text{dist}(x, B) \quad = \quad \|x - P_B(x)\|,$$

where $P_B(x)$ is the projection of $x$ onto $B$. When $B$ is convex, the projection operator $P_B(x)$ is single-valued. For closed nonconvex sets, the projection is multiple-valued for some $x$, but these points are very rare; indeed, they are of Lebesgue measure 0 [22]. The scaled squared distance $\frac{1}{2}$ dist$(x, B)^2$ function is smoother than dist$(x, B)$. One can show that the former function is differentiable with gradient

$$\nabla \frac{1}{2} \text{dist}(x, B)^2 \quad = \quad x - P_B(x) \tag{1}$$

at all points $x$ where $P_B(x)$ is single-valued [23].

The support function $\sigma_A(v) = \max_{x \in A} v^\top x$ and the corresponding support set $\text{supp}_A(v) = \text{argmax}_{x \in A} v^\top x$ also play key roles in our algorithm development. The maximum of $d(A, B)$ exists and necessarily occurs on the boundary of $A$. In fact, Bauer's maximum principle [24,25] implies that the maximum is achieved at an extreme point $x$ of $A$. The point of $B$ corresponding to $x$ occurs on the boundary of $B$, but not necessarily at an extreme point of $B$. The supporting set $\text{supp}_A(v)$ is a singleton if and only if $\sigma_A(v)$ is differentiable at $v$.

Our first algorithm for computing $d(A, B)$,

$$x_{n+1} \in \text{supp}_A(v_n) = \text{supp}_A[x_n - P_B(x_n)] \qquad (2)$$

is a Frank–Wolfe algorithm [26,27]. Our second algorithm,

$$x_{n+1} = P_A(x_n + v_n) = P_A[2x_n - P_B(x_n)] \qquad (3)$$

is a projected gradient algorithm [21,28]. Both algorithms force the objective function $\frac{1}{2} \text{dist}(x, B)^2$ uphill and are iterated until convergence. Because the algorithms can become trapped by local maxima, they are not infallible in finding a global maximum. To overcome this tendency, we introduce a homotopy method that gradually transitions the calculation of the Hausdorff distance from the simple case of two balls to the actual problem of finding $d(A, B)$. Homotopy is one of several heuristics for maximizing multi-modal functions [29]. The crucial difference between projected gradient ascent and the Frank–Wolfe algorithm is that one depends on projection while the other depends on both projection and supporting sets. This difference obviously favors projected gradient ascent.

As a roadmap to the rest of this paper, Section 2 presents (a) the basic notation, (b) a brief overview of the minorization–maximization principle that stands behind the new iterative algorithms (2) and (3), (c) a summary of the support functions and supporting sets, (d) the derivation of both algorithms, (e) a description of the homotopy method, and (f) an explanation of relevant convergence theory. Section 3 tests our two iterative algorithms and the point-cloud method on two representative problems. Both iterative algorithms are orders of magnitude faster than the point-cloud method and benefit from the homotopy heuristic. Projected gradient ascent is also more accurate than the point-cloud method. Section 4 summarizes our conclusions, mentions limitations, and suggests new avenues for research. Appendix A, proves some of the mathematical assertions made in the text and provides the full Julia code for our numerical examples. Note that the code is organized from bottom to top, with the main program occurring at the bottom.

## 2. Materials and Methods

As a prelude to the derivation of the two algorithms, it would be helpful to clarify our notation and make a few remarks about MM algorithms, support functions, and supporting sets. For projected gradient ascent and its homotopy modification, we provide algorithm flowcharts.

### 2.1. Notation

Here are the notational conventions used throughout this article. All vectors appear in boldface. All entries of the vectors $\mathbf{0}$ and $\mathbf{1}$ equal 0 or 1, respectively. The $^\top$ superscript indicates a vector transpose. The Euclidean norm of a vector $x$ is denoted by $\|x\|$. For a smooth real-valued function $f(x)$, we write its gradient (column vector of partial derivatives) as $\nabla f(x)$ and its first differential (row vector of partial derivatives) as $df(x) = \nabla f(x)^\top$. Finally, we denote the directional derivative of $f(x)$ in the direction $v$ by $d_v f(x)$. When $f(x)$ is differentiable, $d_v f(x) = df(x)v$.

### 2.2. MM Algorithms

The algorithms explored here are minorization–maximization (MM) algorithms [23,30]. They depend on surrogate functions $g(x \mid x_n)$ that minorize the original objective $f(x)$ around the current iterate $x_n$ in the sense of satisfying the tangency condition $g(x_n \mid x_n) = f(x_n)$ and the domination condition $g(x \mid x_n) \leq f(x)$ for all $x$. The surrogate balances the two goals of hugging the objective tightly and simplifying maximization. Maximizing the surrogate produces the next iterate $x_{n+1}$ and drives the objective uphill because

$$f(x_{n+1}) \geq g(x_{n+1} \mid x_n) \geq g(x_n \mid x_n) = f(x_n).$$

In minimization, the surrogate majorizes the objective and is instead minimized. The tangency condition remains the same, but the domination condition $g(x \mid x_n) \geq f(x)$ is now reversed. The celebrated EM (expectation–maximization) principle for maximum likelihood estimation with missing data [31] is a special case of minorization–maximization. In the EM setting, Jensen's inequality supplies the surrogate as the expectation of the complete data log-likelihood conditional on the observed data.

### 2.3. Support Functions and Supporting Sets

The set of supporting points $\text{supp}_S(v) = \text{argmax}_{x \in S} v^\top x$ determines the support function $\sigma_S(v)$. For instance, the $\ell_1$ unit ball has $\text{supp}_S(v)$ equal to the convex hull of the vertices $\pm e_i$ where $|v_i|$ is largest. For the unit simplex, $\text{supp}_S(v)$ equals the convex hull of the vertices $e_i$ where $v_i$ is largest. For a Minkowski sum $A + B$, $\text{supp}_{A+B}(v) = \text{supp}_A(v) + \text{supp}_B(v)$. If $S$ is either a convex cone or a compact convex set that is symmetric about the origin with a non-empty interior, then its support function $\sigma_S(y)$ has a special form. In the former case, $\sigma_S(y)$ is the indicator of the dual cone, and in the latter case, $\sigma_S(y)$ is a norm. The support function of a Cartesian product is the Cartesian product of the separate support functions. For instance, the support function of rectangle $[a, b]$ reduces to the one-dimensional case, where $\text{supp}_{[a,b]}(v)$ is $a$ when $v_i < 0$, $b$ when $v_i > 0$, and all of $[a, b]$ when $v_i = 0$. There are many other known support functions. For instance, one-sided penalties such as $c \max\{y, 0\}$ and asymmetric penalties such as $\sigma_{[-a,b]}(y)$ are covered by the current theory. Indeed, the former is the support function generated by the interval $[0, c]$. The latter is the tilted absolute value equal to $by$ for $y \geq 0$ and to $-ay$ for $y < 0$. The support function of a singleton $\{a\}$ is the linear function $a^\top y$. More generally, the support function of the convex hull of the set $\{a_1, \ldots, a_d\}$ is $\max_{1 \leq i \leq d} a_i^\top y$. The support function of the line segment from $-a$ to $a$ is the absolute value $|a^\top y|$. Adding a constant vector $a$ to a set $S$ produces the support function $\sigma_S(y) + a^\top y$. It is trivial to project onto $S + a$ if one can project onto $S$. For any non-negative scalar $c$, the set $cS$ has support function $c\sigma_S(y)$. Again, it is trivial to project onto $cS$ if one can project onto $S$.

### 2.4. Derivation of the Algorithms

When $B$ is convex, the supporting hyperplane inequality

$$\frac{1}{2} \text{dist}(x, B)^2 \geq \frac{1}{2} \text{dist}(x_n, B)^2 + v_n^\top (x - x_n)$$

for $v_n = x_n - P_B(x_n)$ generates our first algorithm. Maximizing this minorization over $x \in A$ is equivalent to calculating the support function $\sigma_A(v_n) = \sup_{x \in A} v_n^\top x$. If $\text{supp}_A(v)$ denotes the set of points in $A$ where $\sigma_A(v)$ is attained, then the Frank–Wolfe algorithm just described can be phrased as

$$x_{n+1} \in \text{supp}_A(v_n) = \text{supp}_A[x_n - P_B(x_n)].$$

The MM principle guarantees that the next iterate $x_{n+1}$ will tend to increase the objective $\frac{1}{2} \text{dist}(x, B)^2$ unless $v_n = 0$. This exception occurs when $x_n \in B$. Fortunately, when the iterates begin in $A \setminus B$, they remain in $A \setminus B$. Indeed, if $x_n \in A \setminus B$ but $x_{n+1} \in B$, then the obtuse angle condition [23] requires

$$
\begin{aligned}
[x_n - P_B(x_n)]^\top x_{n+1} &\leq [x_n - P_B(x_n)]^\top P_B(x_n) \\
&< [x_n - P_B(x_n)]^\top x_n,
\end{aligned}
$$

contradicting the optimality of $x_{n+1}$. To achieve the requirement $x_0 \in A \setminus B$ of the Frank–Wolfe method, we put $x_0 = P_A(r)$ for a random vector $r$ and then check that $\|P_B(x_0) - x\| > 0$.

If $B$ is closed and convex, then the gradient of the function $\frac{1}{2}\operatorname{dist}(x, B)^2$ is Lipschitz with constant 1 [23]. This fact plus the outcome of completing the square entails the minorization

$$
\begin{aligned}
\frac{1}{2}\operatorname{dist}(x, B)^2 &\geq \frac{1}{2}\operatorname{dist}(x_n, B)^2 + v_n^\top (x - x_n) - \frac{1}{2}\|x - x_n\|^2 \\
&= \frac{1}{2}\operatorname{dist}(x_n, B)^2 - \frac{1}{2}\|x - x_n - v_n\|^2 + \frac{1}{2}\|v_n\|^2.
\end{aligned}
$$

Hence, the MM principle implies that defining

$$
x_{n+1} = P_A(x_n + v_n) = P_A[2x_n - P_B(x_n)]
$$

also increases $\frac{1}{2}\operatorname{dist}(x, B)^2$. This second of our two algorithms is a special case of projected gradient ascent. Its flowchart (Algorithm 1) summarizes this straightforward strategy started from many random points $x_0$.

---

**Algorithm 1** Computation of $d(A, B)$ by Projected Gradient Ascent

---

**Require:** Projection operators $P_A(x)$ and $P_B(y)$, initial point $x_0$, maximum iterations $n$, and convergence criterion $\epsilon > 0$.

1: $x = x_0$
2: **for** $iter = 1 : n$ **do**
3:      $x_{\text{new}} = P_A[2x - P_B(x)]$
4:      **if** $\|x_{\text{new}} - x\| < \epsilon$ **then**
5:          break
6:      **else**
7:          $x = x_{\text{new}}$
8:      **end if**
9: **end for**
10: Return $x_{\text{new}}$

---

### 2.5. A Homotopy Method

Although both algorithms are guaranteed to increase the objective, they both suffer from the danger of being trapped by local maxima. One obvious remedy is to launch the algorithms from different random points. A more systematic alternative is to exploit homotopy. The idea is to gradually deform both sets $A$ and $B$ from the unit ball $U$ at the origin, where $d_H(U, U) = 0$ is known, into the target sets $A$ and $B$. In practice, we follow the solution path along the family of set pairs $[tA + (1 - t)U, tB + (1 - t)U]$ from $t = 0$ to $t = 1$. This strategy is viable for projected gradient ascent because we can project points onto the Minkowski convex combination $tC + (1 - t)D$ by three devices. First, it is well known that when $A$ and $B$ are balls with radii $r_A$ and $r_B$ and centers $c_A$ and $c_B$, respectively, the distance $\operatorname{dist}(x, B)$ is maximized by taking $x = c_A - r_A \frac{(c_B - c_A)}{\|c_B - c_A\|}$, unless $A \subset B$, in which case the maximum is 0 [32]. For the convenience of the reader, Proposition A1 of Appendix A proves this assertion. Second, one can exploit the projection identity $P_{tS}(z) = t P_S(t^{-1}z)$ for any $t > 0$. Third, there is an effective algorithm for projecting onto a Minkowski sum $C + D$ [33]. The idea is to alternate the minimization of $\|z - c - d\|$ with respect to $c \in C$ and $d \in D$. The iteration scheme $c_{n+1} = P_C(z - d_n)$ and $d_{n+1} = P_D(z - c_{n+1})$ is guaranteed to converge at a linear rate when either set is strongly convex. Recall that a convex $K$ is strongly convex if there exists an $r > 0$ such that

$$
\alpha x + (1 - \alpha)y + \frac{r}{2}\alpha(1 - \alpha)\|x - y\|^2 z \in K
$$

for all $x$ and $y$ in $K$, $\alpha \in [0, 1]$, and unit vectors $z$ [34]. In particular, $(1 - t)U$ is strongly convex when $t \in [0, 1)$. Furthermore, the Cartesian product of two strongly convex sets is strongly convex [35]. The homotopy method succeeds because the early sets are more

rounded and the objective generates fewer local maxima. The price for better performance is iterations within iterations and an overall slower algorithm. Algorithms 2 and 3 summarize our homotopy strategy for projected gradient ascent.

---

**Algorithm 2** Minkowski Set Projection

---

**Require:** Projection operators $P_A(x)$ and $P_B(y)$, external point $y$, convexity constant $c$, maximum iterations $n$, and convergence criterion $\epsilon > 0$.
1: $a = b = 0$
2: **for** $iter = 1 : n$ **do**
3:      $a_{\text{new}} = cP_A[(y - b)/c]$
4:      $b_{\text{new}} = (1 - c)P_B[(y - a_{\text{new}})/(1 - c)]$
5:      **if** $\|a_{\text{new}} - a\| + \|b_{\text{new}} - b\| < \epsilon$ **then**
6:          break
7:      **else**
8:          $a = a_{\text{new}}$
9:          $b = b_{\text{new}}$
10:      **end if**
11: **end for**
12: Return $a_{\text{new}}$ and $b_{\text{new}}$

---

**Algorithm 3** Homotopy Modification of Projected Gradient Ascent

---

**Require:** Projection operators $P_A(x)$ and $P_B(x)$, centers $c_A$ and $c_B$, and homotopy points $h$.
1: Set $d = \|c_A - c_B\|$ and $x = c_A - \frac{(c_B - c_A)}{d}$        ▷ distance between two balls of radius 1
2: Let $P_U$ be projection onto the unit ball
3: **for** $i = 1 : h - 2$ **do**                  ▷ intermediate homotopy phases
4:      $c = \frac{i}{h-1}$
5:      Put $P_{MA}$ = Minkowski sum projection for $P_A$ and $P_U$ and constant $c$
6:      Put $P_{MB}$ = Minkowski sum projection for $P_B$ and $P_U$ and constant $c$
7:      Perform projected gradient ascent with $P_{MA}$, $P_{MB}$, and initial point $x$
8:      Let $x$ be the outcome
9: **end for**
10: Perform projected gradient ascent with $P_A$ and $P_B$ and initial point $x$
11: Return converged value of $x$          ▷ output final phase of homotopy

---

For the Frank–Wolfe algorithm, similar homotopy tactics apply. For a Minkowski sum $C + D$, $\text{supp}_{C+D}(v) = \text{supp}_C(v) + \text{supp}_D(v)$. This fact plus the identity $\text{supp}_{tC}(v) = \text{supp}_C(tv)$ for $t \geq 0$ makes it possible to carry out the homotopy method.

*2.6. Convergence*

Because this topic has been covered in previous studies [21,36–39], we give an abbreviated treatment here. Each algorithm is summarized by a closed algorithm map $x_{n+1} \in M(x_n)$ that increases the objective $f(x)$. The limit points of the map occur among the stationary points of $f(x)$. By definition, a stationary point $x$ satisfies $d_v f(x) = df(x)v \leq 0$ for all tangent vectors $v$ at $x$. The set of tangent vectors $v$ is the closure of the set of points $c(y - x)$ with $y \in C$ and $c > 0$. This is a place where the convexity of $C$ comes into play. Hence, $x$ is a stationary point if and only if $df(x)x \geq df(x)y$ for all $y \in C$. With this distinction in mind, we state our basic theoretical findings for the Frank–Wolfe method. Homotopy is omitted in these considerations.

**Proposition 1.** *The limit points of the Frank–Wolfe iterates (2) are stationary points of the objective $f(x) = \min_{y \in B} \|x - y\|$ on A. Furthermore, the bound*

$$\min_{0 \le k \le n} \max_{y \in A} df(x_k)(y - x_k) \quad \le \quad \frac{1}{n+1}\left[\max_{x \in A} f(x) - f(x_0)\right]$$

*holds. Thus, the stationary condition* $\max_{y \in A} df(x)(y - x) \le 0$ *is reasonable to expect at a limit point* $x$ *of the Frank–Wolfe algorithm.*

Here is the corresponding finding for projected gradient ascent.

**Proposition 2.** *The limit points of the projected gradient ascent iterates (3) are also stationary points of the objective* $f(x) = \min_{y \in B} \|x - y\|$ *on A. Furthermore, the bound*

$$\min_{0 \le k \le n} \|x_{k+1} - x_k\| \quad \le \quad \sqrt{\frac{2}{(n+1)}\left[f(x_0) - \min_{x \in A} f(x)\right]}$$

*holds.*

Although the convergence rate $O(\frac{1}{\sqrt{n}})$ of projected gradient ascent is slower than the corresponding slow convergence rate $O(\frac{1}{n})$ of the Frank–Wolfe method, in practice, both algorithms usually converge in fewer than 100 iterations. In the case of the Frank–Wolfe algorithm, each iterate is an extreme point. Many convex sets possess only a finite number of extreme points, and convergence to one of them is guaranteed. Unfortunately, the converged point often provides just a local maximum.

### 3. Results

We tested the Frank–Wolfe and projected gradient ascent algorithms on two pairs $(A, B)$ of compact convex sets: (1) where $A$ is the box $[-1, 1]$ translated by $\mathbf{1}$ and $B$ is the intersection of the unit ball and the non-negative orthant; and (2) where $A$ is the probability simplex and $B$ is the $\ell_1$ unit ball translated by $\mathbf{1}$. These examples are representative, and for the second pair one can show that $d_H(A, B) = \sqrt{p}$, where $p$ is the dimension of the ambient space. See Proposition A2 of Appendix A. Table 1 presents our findings. The computation times are in seconds per trial across 100 random initializations and appear to scale affinely (a constant plus linear) in $p$. The columns Maximum, Mean, and Std convey summary statistics of the converged values of the Hausdorff distance. The point-cloud method generated $10^4$ random points to mimic each continuous set. Because the point-cloud method is non-iterative, a single run captured its performance. For the record, all computations were carried out on a MacBook Pro with a 2.3 GHz 8-core i9 chip and 16 GB of memory. Although the algorithms were embarrassingly parallel across trials, our Julia code is completely serial.

The random point-cloud method was not remotely competitive with projected gradient ascent in either accuracy or speed on these sample problems. It did produce approximate distances that confirmed the best results of the iterative methods. As measured by the quality of its solution, projected gradient ascent also outperformed the Frank–Wolfe algorithm. The Frank–Wolfe method was probably too aggressive, perhaps because it moved directly to an extreme point of $A$ in computing $d(A, B)$. On the second problem, projected gradient ascent attained the global maximum across all trials. When the standard deviation of the converged values is positive, it follows that some trials were trapped by inferior local maxima. Both iterative algorithms benefit from the homotopy heuristic, which is fully deterministic. Accordingly, the standard deviations under homotopy equaled 0. Homotopy increased the computation times by less than an order of magnitude for 11 evenly spaced homotopy points. Finally, as $p$ increased, both problems appeared easier to solve by the iterative methods. This behavior was particularly evident when $p = 1000$. In contrast, the random point-cloud solutions deteriorated as $p$ increased.

**Table 1.** Computation of $d_H(A, B)$ by various methods.

| Set Pair | $p$ | Method | Homotopy | Maximum | Mean | Std | Secs |
|---|---|---|---|---|---|---|---|
| (box, ball ∩ orthant) | 2 | proj grad | false | 1.8284 | 1.3314 | 0.40789 | 0.00163 |
| (box, ball ∩ orthant) | 2 | proj grad | true | 1.8284 | 1.8284 | 0.0 | 0.00349 |
| (box, ball ∩ orthant) | 2 | Frank-Wolfe | false | 0.41421 | 0.16569 | 0.20394 | 0.000564 |
| (box, ball ∩ orthant) | 2 | Frank-Wolfe | true | 0.41421 | 0.41421 | 0.0 | 0.00101 |
| (box, ball ∩ orthant) | 2 | point cloud | false | 1.8159 | 1.8159 | 0.0 | 0.625 |
| (simplex, L1 ball) | 2 | proj grad | false | 1.4142 | 1.4142 | 0.0 | 0.001 |
| (simplex, L1 ball) | 2 | proj grad | true | 1.4142 | 1.4142 | 0.0 | 0.00272 |
| (simplex, L1 ball) | 2 | Frank-Wolfe | false | 1.4142 | 1.4142 | 0.0 | 0.000358 |
| (simplex, L1 ball) | 2 | Frank-Wolfe | true | 1.4142 | 1.4142 | 0.0 | 0.0574 |
| (simplex, L1 ball) | 2 | point cloud | false | 1.4142 | 1.4142 | 0.0 | 0.639 |
| (box, ball ∩ orthant) | 3 | proj grad | false | 2.4641 | 1.6358 | 0.50531 | $7.34 \times 10^{-5}$ |
| (box, ball ∩ orthant) | 3 | proj grad | true | 2.4641 | 2.4641 | 0.0 | 0.000605 |
| (box, ball ∩ orthant) | 3 | Frank-Wolfe | false | 0.73205 | 0.31788 | 0.25266 | $7.08 \times 10^{-5}$ |
| (box, ball ∩ orthant) | 3 | Frank-Wolfe | true | 0.73205 | 0.73205 | 0.0 | 0.000145 |
| (box, ball ∩ orthant) | 3 | point cloud | false | 2.4221 | 2.4221 | 0.0 | 0.63 |
| (simplex, L1 ball) | 3 | proj grad | false | 1.7321 | 1.7321 | 0.0 | $5.86 \times 10^{-6}$ |
| (simplex, L1 ball) | 3 | proj grad | true | 1.7321 | 1.7321 | 0.0 | 0.000158 |
| (simplex, L1 ball) | 3 | Frank-Wolfe | false | 1.2247 | 1.2247 | 0.0 | $4.44 \times 10^{-6}$ |
| (simplex, L1 ball) | 3 | Frank-Wolfe | true | 1.2247 | 1.2247 | 0.0 | 0.00186 |
| (simplex, L1 ball) | 3 | point cloud | false | 1.732 | 1.732 | 0.0 | 0.615 |
| (box, ball ∩ orthant) | 10 | proj grad | false | 5.0 | 3.4576 | 0.73165 | 0.0001 |
| (box, ball ∩ orthant) | 10 | proj grad | true | 5.3246 | 5.3246 | 0.0 | 0.000666 |
| (box, ball ∩ orthant) | 10 | Frank-Wolfe | false | 2.0 | 1.2288 | 0.36583 | $9.71 \times 10^{-5}$ |
| (box, ball ∩ orthant) | 10 | Frank-Wolfe | true | 2.1623 | 2.1623 | 0.0 | 0.000183 |
| (box, ball ∩ orthant) | 10 | point cloud | false | 4.8158 | 4.8158 | 0.0 | 0.629 |
| (simplex, L1 ball) | 10 | proj grad | false | 3.1623 | 3.1623 | 0.0 | $7.66 \times 10^{-6}$ |
| (simplex, L1 ball) | 10 | proj grad | true | 3.1623 | 3.1623 | 0.0 | 0.000749 |
| (simplex, L1 ball) | 10 | Frank-Wolfe | false | 2.6667 | 2.6667 | 0.0 | $7.83 \times 10^{-6}$ |
| (simplex, L1 ball) | 10 | Frank-Wolfe | true | 2.6667 | 2.6667 | 0.0 | 0.00196 |
| (simplex, L1 ball) | 10 | point cloud | false | 3.1626 | 3.1626 | 0.0 | 0.613 |
| (box, ball ∩ orthant) | 100 | proj grad | false | 15.248 | 13.016 | 0.69754 | 0.000621 |
| (box, ball ∩ orthant) | 100 | proj grad | true | 19.0 | 19.0 | 0.0 | 0.00126 |
| (box, ball ∩ orthant) | 100 | Frank-Wolfe | false | 7.124 | 6.0078 | 0.34877 | 0.00168 |
| (box, ball ∩ orthant) | 100 | Frank-Wolfe | true | 9.0 | 9.0 | 0.0 | 0.000304 |
| (box, ball ∩ orthant) | 100 | point cloud | false | 15.598 | 15.598 | 0.0 | 0.629 |
| (simplex, L1 ball) | 100 | proj grad | false | 10.0 | 10.0 | 0.0 | $2.77 \times 10^{-5}$ |
| (simplex, L1 ball) | 100 | proj grad | true | 10.0 | 10.0 | 0.0 | 0.000852 |
| (simplex, L1 ball) | 100 | Frank-Wolfe | false | 9.8494 | 9.8494 | 0.0 | $2.21 \times 10^{-5}$ |
| (simplex, L1 ball) | 100 | Frank-Wolfe | true | 9.8494 | 9.8494 | 0.0 | 0.00299 |
| (simplex, L1 ball) | 100 | point cloud | false | 9.9493 | 9.9493 | 0.0 | 0.64 |
| (box, ball ∩ orthant) | 1000 | proj grad | false | 45.174 | 43.698 | 0.6652 | 0.00591 |
| (box, ball ∩ orthant) | 1000 | proj grad | true | 62.246 | 62.246 | 0.0 | 0.00875 |
| (box, ball ∩ orthant) | 1000 | Frank-Wolfe | false | 22.087 | 21.349 | 0.3326 | 0.00256 |
| (box, ball ∩ orthant) | 1000 | Frank-Wolfe | true | 30.623 | 30.623 | 0.0 | 0.00531 |
| (box, ball ∩ orthant) | 1000 | point cloud | false | 48.627 | 48.627 | 0.0 | 1.13 |
| (simplex, L1 ball) | 1000 | proj grad | false | 31.623 | 31.623 | 0.0 | 0.000377 |
| (simplex, L1 ball) | 1000 | proj grad | true | 31.623 | 31.623 | 0.0 | 0.0113 |
| (simplex, L1 ball) | 1000 | Frank-Wolfe | false | 31.575 | 31.575 | 0.0 | 0.000303 |
| (simplex, L1 ball) | 1000 | Frank-Wolfe | true | 31.575 | 31.575 | 0.0 | 0.013 |
| (simplex, L1 ball) | 1000 | point cloud | false | 31.597 | 31.597 | 0.0 | 1.11 |

## 4. Discussion

The Hausdorff distance problem is intrinsically interesting, with theoretical applications throughout mathematics and practical applications in image processing. Given the non-convexity of the problem, it has not received nearly the attention in the mathematical literature as the closest point problem. Exact values of $d_H(A, B)$ are available in a few special cases such as the two highlighted in our appendix. Research on fast algorithms tends

to be limited to random point clouds. Infinite sets defined by mathematical formulas have been largely ignored. The current paper partially rectifies this omission and demonstrates the value of continuous optimization techniques. The Frank–Wolfe and projected gradient ascent algorithms are relatively easy to code and extremely fast, even in high dimensions. Our preliminary experiments tilt toward projected gradient ascent as the more reliable of the two options. A naive implementation of the point-cloud method is not remotely competitive with projected gradient ascent. More exhaustive studies are warranted beyond the proof of principle examples presented here.

The standard convergence arguments covered in Section 2.6 guarantee that all limit points of the two algorithm classes are stationary points. In practice, convergence appears much faster than the slow rates mentioned in Propositions 1 and 2. We suspect, but have not proved, that full convergence to a stationary point always occurs. This exercise would require a foray into the difficult terrain of real algebraic geometry [40]. In any event, convergence to a global maximum is not guaranteed. Fortunately, safeguards can be put in place to improve the chances of successful convergence. The homotopy method capitalizes on the exact distance between two balls. Minkowski set rounding smooths the boundary of the target sets and steers iterates in a productive direction.

The computation of the Hausdorff distance $d_H(A, B)$ is apt to be much more challenging when either $A$ or $B$ is non-convex. Many sets can be represented as finite unions of compact convex sets. If $A = \cup_i A_i$ and $B = \cup_j B_j$, then the computation of $d_H(A, B)$ reduces to the computation of $d(A_i, B)$ for each index $i$ and $d(B_j, A)$ for each index $j$. The identities $d(A, B) = \max_i d(A_i, B)$ and $d(B, A) = \max_i d(B_j, A)$ make this claim obvious. The further identity $\mathrm{dist}(x, B) = \min_j \mathrm{dist}(x, B_j)$ implies that

$$d(A, B) \quad = \quad \max_i d(A_i, B) \quad = \quad \max_i \max_{x \in A_i} \min_j \mathrm{dist}(x, B_j).$$

In general, saddlepoint problems of this sort are hard to solve. One possible line of attack is to minorize $\min_j \mathrm{dist}(x, B_j)$ and then maximize the minorization.

The validation and implementation of this strategy will require substantial effort beyond the introduction to the problem pursued in the current paper. Let us merely add that the fast implementation of a more general Hausdorff distance algorithm will depend on the nature of the candidate sets $A_i$ and $B_j$. In the plane, triangles are appealing [41,42]. It is straightforward to project onto a triangle, and a triangle by definition possesses exactly three extreme points. Furthermore, a great deal of research under the heading of finite elements has identified good algorithms for triangulating complicated regions of the plane. The software Triangle for generating two-dimensional meshes and Delaunay triangulations is surely pertinent [43]. The triangularization of surfaces forms part of the MESH software for Hausdorff distance estimation [44].

We hope this paper will provoke a greater focus on the Hausdorff distance problem. As a prototype non-convex problem, it is worthy of far more attention. Continuous optimization tools can be brought to bear on the problem and may ultimately generate more efficient algorithms than the discrete algorithms designed for finite point clouds. The fact that our algorithms scale well in higher dimensions is a plus. We would also like to highlight the illumination that the MM principle brings to the construction of new high-dimensional optimization algorithms, including those considered here. Although neglected in the past, MM may well be the single most unifying principle of algorithm construction in continuous optimization.

**Funding:** Research supported in part by USPHS grants GM53275 and HG006139.

**Data Availability Statement:** Not applicable.

**Conflicts of Interest:** The author declares no conflict of interest.

**Appendix A**

*Appendix A.1. Proofs*

**Proposition A1.** *If A and B are balls with radii $r_A$ and $r_B$ and centers $c_A$ and $c_B$, respectively, then $\mathrm{dist}(x, B) = \|x - c_B\| - r_B$ for $x \notin B$, and $d(A, B) = \|c_A - c_B\| + r_A - r_B$, unless $A \subset B$, in which case the maximum is 0.*

**Proof.** The first assertion is obvious. To maximize $\mathrm{dist}(x, B)$ over $A$, we form the Lagrangian

$$\mathcal{L}(x, \lambda) = -\frac{1}{2}\|x - c_B\|^2 + \frac{1}{2}\lambda(\|x - c_A\|^2 - r_A^2).$$

The stationary condition

$$0 = -x + c_B + \lambda(x - c_A)$$

implies that

$$x - c_A = \frac{c_B - \lambda c_A}{1 - \lambda} - c_A = \frac{(c_B - c_A)}{1 - \lambda}$$

and determines $\lambda$ through

$$r_A^2 = \frac{\|c_B - c_A\|^2}{(1 - \lambda)^2}.$$

It follows that

$$x = c_A \pm r_A \frac{(c_B - c_A)}{\|c_B - c_A\|}$$

and that

$$\begin{aligned}
\mathrm{dist}(x, B) &= \|x - c_B\| - r_B \\
&= \left\|c_A - c_B \pm r_A \frac{(c_B - c_A)}{\|c_B - c_A\|}\right\| - r_B \\
&= \left|1 \mp \frac{r_A}{\|c_A - c_B\|}\right| \|c_A - c_B\| - r_B.
\end{aligned}$$

Geometrically, $\max_{x \in A} \mathrm{dist}(x, B) = \|c_A - c_B\| + r_A - r_B$ should hold, so $x = c_A - r_A \frac{(c_B - c_A)}{\|c_B - c_A\|}$ gives the correct sign.  $\square$

**Proposition A2.** *If A is the probability simplex in $\mathbb{R}^p$ and B is the $\ell_1$ ball U translated by $\mathbf{1}$, then $d_H(A, B) = \sqrt{p}$.*

**Proof.** Consider first $d(A, B)$. The maximum of $d(x, B)$ occurs at an extreme point of $A$, say $x = (1, 0, \ldots, 0)^\top = e_1$ by symmetry. On $U$ the convex function

$$f(y) = \|y + \mathbf{1} - e_1\|^2 = y_1^2 + \sum_{i>1}(y_i + 1)^2$$

achieves its minimum value when all $y_i$ are equal for $i > 1$. The common value $z$ should satisfy $z \leq 0$, while $y_1$ can have either sign. For $y_1 \in [0, 1]$, we accordingly minimize $y_1^2 + (p - 1)(z + 1)^2$ subject to $y_1 - (p - 1)z \leq 1$. We can decrease $z$ until $y_1 - (p - 1)z = 1$ and solve for $z = \frac{y_1 - 1}{p - 1}$. Thus, we must minimize $q(y_1) = y_1^2 + (p - 1)\frac{(y_1 + p - 2)^2}{(p - 1)^2}$ over $[0, 1]$. Now the stationary point $-\frac{p - 2}{p}$ of $q(y_1)$ falls outside $[0, 1]$, so the minimum occurs

at either 0 or 1. Because $q(0) = \frac{(p-2)^2}{(p-1)}$ and $q(1) = 1 + p - 1 = p$, the point 0 wins, and $d(A, B) = \sqrt{\frac{(p-2)^2}{p-1}}$.

Next, consider $d(B, A)$. The maximum of $d(y, A)$ occurs at an extreme point of $B$, say $y = \pm e_1 + \mathbf{1}$ by symmetry. On $A$ the convex function

$$f(x) = \|x - \mp e_1 - \mathbf{1}\|^2 = (x_1 \mp 1 - 1)^2 + \sum_{i>1}(x_i - 1)^2$$

is maximized by taking $y = e_1 + \mathbf{1}$. We make this choice and again assume $x_i = z$ for $i > 1$. Now $z \geq 0$, and we increase $z$ until $x_1 + (p-1)z = 1$. This gives $z = \frac{1-x_1}{p-1}$ and reduces $f(x)$ to the quadratic

$$q(x_1) = x_1^2 + (p-1)\frac{(2-p-x_1)^2}{(p-1)^2}.$$

Now the stationary point $-\frac{p-2}{p}$ of $q(x_1)$ falls outside $[0, 1]$, so the minimum occurs at either 0 or 1. Thus, $q(0) = \frac{(p-2)^2}{p-1}$ and $q(1) = 1 + p - 1 = p$, and the point 1 wins. Finally,

$$d_H(A, B) = \max\left\{\sqrt{\frac{(p-2)^2}{(p-1)}}, \sqrt{p}\right\} = \sqrt{p}. \quad \square$$

*Appendix A.2. Julia Computer Code*

```julia
using LinearAlgebra, Distances, Random, StatsBase

"""Generates a random point in the box [a,b]."""
function RandomBox(a, b)
  n = length(a)
  return a + rand(n) .* (b - a)
end

"""Generates a random point in a Euclidean ball."""
function RandomBall(radius, center)
  n = length(center)
  x = randn(n)
  x = x / norm(x)
  r = rand()^(1 / n)
  return (radius * r) * x + center
end

"""Generates a random point in the intersection of a ball centered
at the origin and the nonnegative orthant."""
function RandomBallOrthant(radius, n)
  x = RandomBall(radius, zeros(n))
  return abs.(x)
end

"""Generates a random point in the probability simplex."""
function RandomSimplex(n)
  x = -log.(rand(n))
  return x / sum(x)
end

"""Generates a random point in an L1 ball."""
function RandomL1Ball(radius, center)
  n = length(center)
  x = -log.(rand(n))
  x = x / sum(x)
  for i = 1:n
    if rand() < 1 / 2
      x[i] = - x[i]
    end
  end
```

```julia
    r = rand()^(1 / n)
    return (radius * r) * x + center
end

"""Computes the Hausdorff distance between the point sets A and B."""
function hausdorff(A, B)
  D = pairwise(Euclidean(), A, B)
  dAB = maximum(minimum(D, dims = 2))
  dBA = maximum(minimum(D, dims = 1))
  return max(dAB, dBA)
end

"""Projects the point y onto a re-centered set."""
function RecenterProjection(Proj, y::Vector{T}, c::Vector{T}) where T <: Real
  return Proj(y - c) + c # set is translated by c
end

"""Projects the point y onto a scaled set."""
function ScaleProjection(Proj, y::Vector{T}, s::T) where T <: Real
  return s * Proj(y / s) # s > 0 is the scaling factor
end

"""Projects the point y onto the closed ball with radius r."""
function BallProjection(y::Vector{T}, r = one(T)) where T <: Real
#
  distance = norm(y)
  if distance > r
    return (r / distance) * y
  else
    return y
  end
end

"""Projects the point y onto the closed box with bounds a and b."""
function BoxProjection(y::Vector{T}, a = -ones(T, length(y)),
  b = ones(T, length(y))) where T <: Real
#
  return clamp.(y, a, b)
end

"""Projects the point y onto the simplex {x | x >= 0, sum(x) = r}."""
function SimplexProjection(y::Vector{T}, r = one(T)) where T <: Real
#
  n = length(y)
  z = sort(y, rev = true)
  (s, lambda) = (zero(T), zero(T))
  for i = 1:n
    s = s + z[i]
    lambda = (s - r) / i
    if i < n && lambda < z[i] && lambda >= z[i + 1]
      break
    end
  end
  return max.(y .- lambda, zero(T))
end

"""Projects the point y onto the ell_1 ball with radius r."""
function L1BallProjection(y::Vector{T}, r = one(T)) where T <: Real
#
  p = abs.(y)
  if norm(p, 1) <= r
    return y
  else
    x = SimplexProjection(p, r)
    return sign.(y) .* x
  end
end

"""Projects the point y onto the intersection of the ball of
```

```
                            radius r and the nonnegative orthant."""
                            function BallAndOrthantProjection(y::Vector{T}, r = one(T)) where T <: Real
                            #
                              x = copy(y)
                              x = max.(x, zero(T)) # project onto orthant
                              return (r / max(norm(x), r)) .* x
                            end

                            """Finds the support point for y on the inflated unit ball."""
                            function BallSupp(y::Vector{T}, r = one(T)) where T <: Real
                            #
                              return (r / norm(y)) * y
                            end

                            """Finds the support point for y on the box [a, b]."""
                            function BoxSupp(y::Vector{T}, a = -ones(T, length(y)),
                              b = ones(T, length(y))) where T <: Real
                            #
                              n = length(y)
                              x = zeros(T, n)
                              for i = 1:n
                                if y[i] > zero(T)
                                  x[i] = b[i]
                                elseif y[i] < zero(T)
                                  x[i] = a[i]
                                else
                                  x[i] = (a[i] + b[i]) / 2
                                end
                              end
                              return x
                            end

                            """Finds the support point for y on the simplex {x | x >= 0, sum(x) = r}."""
                            function SimplexSupp(y::Vector{T}, r = one(T)) where T <: Real
                            #
                              x = zeros(T, length(y))
                              (v, m) = findmax(y)
                              x[m] = r
                              return x
                            end

                            """Finds the support point for y on the L1 ball."""
                            function L1BallSupp(y::Vector{T}, r = one(T)) where T <: Real
                            #
                              x = zeros(T, length(y))
                              (v, m) = findmax(abs, y)
                              x[m] = sign(y[m]) * r
                              return x
                            end

                            """Finds the support point for y on the intersection of the ball of
                            radius r and the nonnegative orthant."""
                            function BallAndOrthantSupp(y::Vector{T}, r = one(T)) where T <: Real
                            #
                              x = max.(y, zero(T))
                              if sum(x) <= zero(T)
                                return zeros(T, length(y))
                              else
                                return (r / norm(x)) * x
                              end
                            end

                            """Projects the point y onto the Minkowski rounded set
                            R = c * S + (1 - c) * B. Here B is the unit ball, Proj
                            is projection onto S, and Proj_R(y) = a + b."""
                            function MinkowskiNear(Proj, y, c, conv)
                            #
                              n = length(y)
                              (aold, bold) = (zeros(n), zeros(n))
```

```
    (anew, bnew) = (zeros(n), zeros(n))
    for iter = 1:100
      anew = c .* Proj((y - bold) ./ c) # project onto c * S
      bnew = (1 - c) .* BallProjection((y - anew) ./ (1 - c))
      if norm(aold - anew) + norm(bold - bnew) < conv
        break
      else
        @. aold = anew
        @. bold = bnew
      end
    end
    return anew + bnew
end

"""Finds the farthest point on A from B by Frank-Wolfe."""
function FrankWolfe(SuppA, PB, x0)
  (xold, xnew) = (copy(x0), similar(x0))
  for iter = 1:100
    xnew = SuppA(xold - PB(xold))
    if norm(xnew - xold) < 1.0e-10
      break
    else
      xold .= xnew
    end
  end
  far = norm(xnew - PB(xnew))
  return (far, xnew)
end

"""Finds the farthest point on A from B by projected gradient ascent."""
function farthest(PA, PB, x0)
  (xold, xnew) = (copy(x0),copy(x0))
  for iter = 1:100
    xnew = PA(2xold - PB(xold))
    if norm(xnew - xold) < 1.0e-10
      break
    else
      xold .= xnew
    end
  end
  far = norm(xnew - PB(xnew))
  return (far, xnew)
end

"""Finds the farthest point on A from B by homotopy."""
function farthest_homotopy(PA, PB, SA, CenterA, CenterB, x0, n, method)
  x = BallProjection(x0)
  (far, homotopy_points, conv) = (0.0, 10, 1.0e-10)
  for iter = 0:homotopy_points
    if iter == 0 # ball to ball
      d = norm(CenterA - CenterB)
      (far, x) = (d, (1 + 1 / d) * CenterA - CenterB / d)
    elseif iter == homotopy_points # d(A, B)
      if method == "proj grad"
        (far, x) = farthest(PA, PB, x)
      elseif method == "Frank-Wolfe"
        (far, x) = FrankWolfe(SA, PB, x)
      end
    else # intermediate case
      t = iter / homotopy_points
      PMB(z) = MinkowskiNear(PB, z, t, conv)
      if method == "proj grad"
        PMA(z) = MinkowskiNear(PA, z, t, conv)
        (far, x) = farthest(PMA, PMB, x)
      elseif method == "Frank-Wolfe"
        SM(z) = SA(t * z) + BallSupp((1 - t) * z)
        (far, x) = FrankWolfe(SM, PMB, x)
      end
    end
```

```
      end
    return (far, x)
  end

  """Orchestrates Hausdorff distance estimation."""
  function master(ProjA, ProjB, SuppA, SuppB, CenterA, CenterB, method,
    homotopy, n, trials, io)
  #
    (count, tries, optimum, obj) = (0, 100, 0.0, zeros(trials))
    x0 = zeros(n)
    PA(z) = RecenterProjection(ProjA, z, CenterA)
    PB(z) = RecenterProjection(ProjB, z, CenterB)
    for trial = 1:trials
      success = false
      for i = 1:tries # find a point in A \ B
        x0 = PA(randn(n))
        if norm(PB(x0) - x0) > 1.0e-10
          success = true
          break
        end
      end
      if homotopy # solve for d(A, B)
        (objA, xA) = farthest_homotopy(PA, PB, SuppA, CenterA, CenterB,
          x0, n, method)
      else
        if method == "proj grad" && success # solve for d(A, B)
          (objA, xA) = farthest(PA, PB, x0)
        elseif method == "Frank-Wolfe" && success
          (objA, xA) = FrankWolfe(SuppA, PB, x0)
        else
          objA = 0.0
        end
      end
      success = false
      for i = 1:tries # find point in B \ A
        x0 = PB(randn(n))
        if norm(PA(x0) - x0) > 1.0e-10
        success = true
          break
        end
      end
      if homotopy # solve for d(B, A)
        (objB, xB) = farthest_homotopy(PB, PA, SuppB, CenterB, CenterA,
          x0, n, method)
      else
        if method == "proj grad" && success
          (objB, xB,) = farthest(PB, PA, x0)
        elseif method == "Frank-Wolfe" && success
          (objB, xB) = FrankWolfe(SuppB, PA, x0)
        else
          objB = 0.0
        end
      end
      obj[trial] = max(objA, objB) # Hausdorff distance
      if obj[trial] > optimum + 10.0e-10 # update count of maximum distance
        count = 1
        optimum = obj[trial]
      elseif obj[trial] > optimum - 10.0e-8
        count = count + 1
      end
    end
    (avg, stdev) = (mean(obj), std(obj))
    if stdev < 1.0e-10 stdev = 0.0 end
    return (fraction = count / trials, optimum, avg, stdev)
  end

outfile = "Hausdorff.out";
io = open(outfile, "w");
trials = 100;
```

```
points = 10000
println(io,"Set Pair"," & ","p"," & ","Method"," & ","Homotopy"," & ",
  "Maximum"," & ","Mean"," & ","Std"," &  ","Seconds"," \\ ")
for n in [2, 3, 10, 100, 1000]
  for i = 1:2
    if i == 1
      CenterA = ones(n)
      CenterB = zeros(n)
      ProjA = BoxProjection
      ProjB = BallAndOrthantProjection
      SuppA = BoxSupp
      SuppB = BallAndOrthantSupp
      title = "dH(box, ball and orthant)"
    elseif i == 2
      CenterA = zeros(n)
      CenterB = ones(n)
      ProjA = SimplexProjection
      ProjB = L1BallProjection
      SuppA = SimplexSupp
      SuppB = L1BallSupp
      title = "dH(simplex, L1 ball)"
    end
#
    (method, homotopy) = ("proj grad", false);
    Random.seed!(1234)
    time = @elapsed (fraction, optimum, avg, stdev) = master(ProjA,
      ProjB, SuppA, SuppB, CenterA, CenterB, method, homotopy, n,
      trials, io)
    println(io,title," & ",n," & ",method," & ",homotopy," & ",
        round(optimum, sigdigits=5)," & ",round(avg, sigdigits=5)," & ",
        round(stdev, sigdigits=5)," & ",round(time/trials, sigdigits=3)," \\ ")
#
    (method, homotopy) = ("proj grad", true);
    Random.seed!(1234)
    time = @elapsed (fraction, optimum, avg, stdev) = master(ProjA,
      ProjB, SuppA, SuppB, CenterA, CenterB, method, homotopy, n,
      trials, io)
    println(io,title," & ",n," & ",method," & ",homotopy," & ",
      round(optimum, sigdigits=5)," & ",round(avg, sigdigits=5)," & ",
      round(stdev, sigdigits=5)," & ",round(time/trials,sigdigits=3)," \\ ")
#
    (method, homotopy) = ("Frank-Wolfe", false);
    Random.seed!(1234)
    time = @elapsed (fraction, optimum, avg, stdev) = master(ProjA,
      ProjB, SuppA, SuppB, CenterA, CenterB, method, homotopy, n,
      trials, io)
    println(io,title," & ",n," & ",method," & ",homotopy," & ",
      round(optimum, sigdigits=5)," & ",round(avg, sigdigits=5)," & ",
      round(stdev, sigdigits=5)," & ",round(time/trials,sigdigits=3)," \\ ")
#
    (method, homotopy) = ("Frank-Wolfe", true);
    Random.seed!(1234)
    time = @elapsed (fraction, optimum, avg, stdev) = master(ProjA,
      ProjB, SuppA, SuppB, CenterA, CenterB, method, homotopy, n,
      trials, io)
    println(io,title," & ",n," & ",method," & ",homotopy," & ",
      round(optimum, sigdigits=5)," & ",round(avg, sigdigits=5)," & ",
      round(stdev, sigdigits=5)," & ",round(time/trials,sigdigits=3)," \\ ")
#
    (method, homotopy) = ("point cloud", false);
    Random.seed!(1234)
    points = 10000
    A = zeros(n, points)
    B = zeros(n, points)
    if i == 1
      (a, b) = (ones(n), 2 * ones(n))
      for j = 1:points
        A[:, j] = RandomBox(a, b)
        B[:, j] = RandomBallOrthant(1.0, n)
```

```
          end
        else
          for j = 1:points
            A[:, j] = RandomSimplex(n)
            B[:, j] = RandomL1Ball(1.0, ones(n))
          end
        end
        time = @elapsed optimum = hausdorff(A, B)
        println(io,title," & ",n," & ",method," & ",homotopy," & ",
          round(optimum, sigdigits=5)," & ",round(optimum, sigdigits=5)," & ",
          round(0.0, sigdigits=5)," & ",round(time, sigdigits=3)," \\ ")
      end
    end
    close(io)
```

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
