# Peer review of "Computation of the Hausdorff Distance between Two Compact Convex Sets"

_algorithms, doi:10.3390/a16100471_

Round 1

Reviewer 1 Report

The authors have investigated the Hausdortt distance between two compact convex sets. The Frank-Wolfe and projected gradient algorithms have been employed to test on two pairs (A, B) of compact convex sets. This study is interesting however there are some drawbacks that the authors should address them to improve this study.

  1. The main contribution of this study must be described clearly in the introduction.
  2. The authors must state the scientific basis of using Hausdorff distance in this study. What is the prioritization of the proposed method with the Euclidean distance before.
  3. In the results, the authors only provided table 1. What is the conclusion remark from this result.
  4. The discussion is inadequate to decide the proposed method is more effective than the traditional methods.
  5. The acknowledgement should be implemented in this study.

The writing in English should be improved.

Author Response

I thank the reviewer for his/her helpful comments. My manuscript has been extensively revised accordingly. New or revised verbiage is highlighted in red. Eight references have been added. Comparison with the most popular competing method has been inserted in Table 1, which is now better explained. My responses to the numbered queries are:

  1. The introduction clearly states two new generic algorithms for computing Hausdorff distances.
  2. The additional references reinforce the importance of Hausdorff distance in the scientific literature.
  3. I demonstrate that the projected gradient ascent outperforms the currently most popular competing method.
  4. The Results and Discussion sections now clearly state my conclusions. Table 1 is expanded and better explained.
  5. This is a solo paper with no acknowledgments.

Reviewer 2 Report

The paper proposed for publication constructed Frank-Wolfe and projected gradient ascent algorithms for computing the Hausdorff distance between two compact convex sets. 

Test sets were prepared and modified to calculate this distance.

Comments:

1. Introducion should describe what applications existing solutions have and how the Hausdorff distance is calculated. Please remove the sentence from line 37 page 1. Do not start with the introduction formulas, but go to the formulas after a solid description of the existing methods and algorithms.

2. All quantity inputs (even x, y) should be explained.

3. Information from line 61 on page 2 should be in the Methods chapter.

4. line 73 page 2 MM algorithms??? tu powinno być wyjaÅ›nienie skrótu zamiast line 86 page 3.

5. The Methods chapter should include a flowchart of the algorithm. Because the linear algorithm is in Julia code - and it should be a diagram so that it is recognizable to other programming languages.

6. Formulas are not numbered (line 26-27 page 1, line 49 page 2, line 91 and 120 page 3 and others).

7. Table 1. Performance of Projected Gradient and Frank–Wolfe Algorithms - should be describe.

8. Is it possible to visualize the sets and calculation results taken for analysis?

9. The results should be compared with existing method for Hausdorff distance calculate.

10.. Author Contributions, Funding, Data Availability Statement Conflicts of Interest - are missing.

Author Response

I thank the reviewer for his/her helpful comments. My manuscript has been accordingly revised. The minimum word count is now exceeded. New or revised verbiage is highlighted in red. Eight references have been added. Comparison with the most popular competing method has been inserted in Table 1, which is now better explained. My responses to the numbered queries are:

1. The added references give a better idea of the importance of Hausdorff distance and its applications. As recommened,
some explanatory material has been moved to the Methods section. The basic formulas for the new Franke-Wolfe and projected gradient ascent algorithms has been retained in the Introduction. I now emphasize comparisons with the finite point-cloud method, the most commonly applied method in practice.

2. My notational lapses have been corrected.

3. Most of the material on the homotopy method has been moved to the Methods section.

4. I don't understand this query.

5. Flowcharts are now provided.

6. My convention is to only number equations when they are subsequently referenced. This avoids clutter.

7. Table 1 has been expanded and is now better described.

8. The sets in question are well documented and depicted in the literature. I have added Blaschke's formula for Hausdorff distance. This helps in understanding.

9. I now compare my methods to the finite point-cloud method. It would be a major undertaking to compare across the board. Given the 10 day revision period, this is impossible.

10 Funding is listed on the title page. The data is simulated, and I hope readers will have access to my Julia code. No conflicts of interest exist, and author contributions are unnecessary for a solo paper.

Reviewer 3 Report

(1)Please check the manuscript carefully. d(x,B), dis(y,A). Why did the authors use two different function names (d, dis)?

(2)Most references were published early. Are there any more related works published recently? Can the author compare his/her algorithm with the related works?

(3)The author claimed that his/her algorithm can avoid being trapped by local maxima. How to confirm this advantage? Theoretical analysis and experiments?

(4)In addition to Table 1, can the author show more experiments and performance? E.g. computational complexity.

None.

Author Response

I thank the reviewer for his/her helpful comments. My manuscript has been accordingly revised. The minimum word count is now exceeded. New or revised verbiage is highlighted in red. Eight references have been added. Comparison with the most popular competing method has been inserted in Table 1, which is now better explained. My responses to the numbered queries are:

  1. The inconsistencies in notation have been fixed.
  2. Some of the added references are newer. Please keep in mind that this is an old problem. My algorithms arise from a new perspective. I now compare my results to the finite point-cloud method. It would be a major undertaking to compare across the board. Given the 10 day revision period, this is impossible.
  3. Trapping occurs whenever the standard deviation of the computed distances over many random trials is positive. One of the two problems has an exact solution, as I demonstrate in the appendix. The approximate results from point-cloud method tend to confirm the best solutions of the new iterative methods.
  4. I mention the computational complexity of the point-cloud method. The computational complexity of projected gradient ascent depends on the computational complexity of the projection operator, so no general result is possible. However, the examples illustrate that the speed of the algorithms can be impressive even in high dimensions.

Round 2

Reviewer 1 Report

The authors have addressed some comments from this reviewer. This manuscript is better however there are some typing mistakes which have been found in this paper.

Kindly recheck all main text carefully before resubmitting to Journal.

Author Response

I can not the find typos mentioned in your review. If you are specific and mention section, paragraph number, and context, then I will gladly fix the typos.

Reviewer 2 Report

I am very sorry, but in my opinion the amendments introduced are not sufficient.

The flowchart should be in the form of a drawing (a flowchart, not a linear algorithm) so that it is recognizable in other programming languages.

point 4 was in my native language.

point 4:  now line 92 MM algorithms??? there should be an explanation of the abbreviation here instead line 116.

Author Response

Thanks for your further comments. I have replaced the acronym "MM" by the more specific "minorization-maximization principle".  The principle is explained in Section 2.2 shortly after its first reference.

There is a great deal of background material, including two-dimensional illustrations, on the web about Hausdorff distance. I now specifically mention the Wikipedia entry on Hausdorff distance.

As for your concern about flowcharts, which I take to be equivalent to algorithm descriptions,  Algorithms 1 and 2 should are easy to program in any language that allows one to pass functions into functions. I have avoided making my paper an advertisement for Julia, which has this capacity. Algorithm 3 is admittedly terse and relies on this strength of Julia. For these reasons I include my Julia code in the appendix. A serious programmer can see all the details there, including the required projection operators and support set functions. The Julia code is internally documented and relies on good variable names to improve readability.  In any case, I don't see how to improve the algorithm descriptions without getting enmeshed in a morass of detail. If you have additional advice, please be as specific as possible.

Reviewer 3 Report

Accept.

None.

Author Response

I thank reviewer # 3 for accepting my revision.